# Can UNESCO Use Blockchain to Ensure the Intangible Cultural Heritage of Humanity? A Systemic Approach That Explains the Why, How, and Difficulties of Such a Venture

**Nikolaos Zoannos** †, **Pelagia Chourdaki** *,† **and Nikitas Assimakopoulos**

Ph.D. Candidate, Hellenic Society for Systemic Studies (HSSS), University of Piraeus, 11851 Athina, Greece; nmzoanno@unipi.gr (N.Z.); assinik@unipi.gr (N.A.)
* Correspondence: pchourdaki@unipi.gr; Tel.: +30-6977321430
† These authors contributed equally to this work.

**Abstract:** Focusing on the implementation of the 2003 UNESCO Convention for the Safeguarding of the Intangible Cultural Heritage (ICH), this article highlights the fact that the cataloging procedures, the way records are stored, and the metadata models used during the recording, visualization, and storage of ICH expression elements, vary from country to country. Especially in these days of great uncertainty, war conflicts, and systemic risks that may arise, it is vital to clarify what techniques will be used in the storage of ICH to ensure its unhindered preservation and dissemination over time across the globe. Using the systems thinking approach indicated for cases of great complexity, the process followed in Greece for depositing a new element in its local repository is described to demonstrate the need for a well-defined process by UNESCO, which must be followed worldwide, and which has not been defined so far. What are the potential challenges not only in determining the type of data, but also in choosing the best metadata model to use in each case when capturing these data? What technologies must be used for storing digital cultural heritage in such a way that will preserve it over time, defying physical and technological hazards? This article finally suggests how blockchain technologies (BT) can be effectively used to store the global ICH and ensure its continuity in future generations by creating a decentralized worldwide network between the heritage stakeholders.

**Keywords:** UNESCO; blockchain technologies (BT); decentralized (DLT); intangible cultural heritage (ICH); digital management; metadata models; data schemes; systems thinking; systemic approach; systemic risks



## 1. Introduction

Intangible cultural heritage (ICH) is a living heritage that declares humanity's cultural diversity and its maintenance plays a crucial role in defining the cultural identity of each country in the face of growing globalization [1]. In the past few years, a remarkable effort by UNESCO has been recorded for the preservation (capturing, visualizing and presenting) of the world's ICH, that is the traditions or the living expressions which are transmitted from the previous generations to the next ones. ICH is traditional, contemporary, and living at the same time. It is a living cultural phenomenon that changes, evolves, and passes on to the next generations. Its importance is crucial because it contributes to social cohesion, encouraging a sense of identity and responsibility, and thus helping individuals to feel part of one or different communities and to feel part of society at large. It is very important to underline that ICH is community-based. Only the communities, groups, or individuals that create, maintain, and transmit ICH can recognize and decide that a given expression or practice is their heritage.

The milestone, when someone refers to ICH management, is the 2003 Convention for the Safeguarding of the ICH of UNESCO [2]. It is an international treaty that has currently

been ratified, approved, or accepted by 194 State Parties (May 2022). The Convention provides an international framework for planning actions to preserve and promote cultural expressions (708 elements corresponding to 140 countries, December 2022) [3] with no tangible material dimension and proposes 5 domains: (i) oral tradition and expressions, including language as a vehicle of ICH (fairy tales, myths, and narratives, narrative songs), (ii) performing arts (vocal and instrumental music, dance, folk theater), (iii) social practices and festive events (folk dances, customs practiced on annual basis, important stages in human life), (iv) knowledge and practices concerning nature and universe (traditional cultivation practices, ethnobotanical knowledge, popular perceptions of meteorology, etc.), and (iv) traditional craftsmanship (the know-how associated with traditional craftsmanship as weaving, pottery, woodworking, etc.).

The implementation of the Convention is determined by an institutional and a legal framework. It operates at a global and national level.

The safeguarding of ICH at an international level is carried out through two lists: the "Representative List of the Intangible Cultural Heritage of Humanity" and the "List of Intangible Cultural Heritage in Need of Urgent Safeguarding" (if there is an immediate danger of extinction for this element), and in addition by the "Register of good safeguarding practices of the Intangible Cultural Heritage" [4]. These two lists act as mechanisms for identifying traditional expressions of ICH on an international level. The lists are accessible online and they are indeed a relatively limited repository that includes photographs and audio-visual recordings and raises awareness about the listed elements and their communities.

The convention also draws up national inventories where the heritage compilations of each State Party are recorded. Subsequently, some of the native cultural expressions from the national inventories are qualified under standardized procedures and registered in the international UNESCO lists.

However, the terms "Lists", Register", and "Inventories" are a naming convention that the States Parties are strongly advised to follow in order to avoid confusion between the different mechanisms of listing at national and international levels [5] (p. 6).

Each member state of the 2003 Convention uses its own process for inscribing a new element on UNESCO's lists. Although we have seen many countries such as Greece, China, France, Croatia, Mauritania, Iran, Brazil, United Arab Emirates, Uganda, Morocco, Egypt, and many more which have successfully followed the procedure to add a new element to their intangible cultural heritage, it is worth highlighting that only 76 of those 708 elements, belonging in 40 different countries, have been added since 2009 in the List of Intangible Cultural Heritage in Need of Urgent Safeguarding [6].

More specifically, 12 elements were added in 2009, 4 elements in 2010, 10 elements in 2011, 4 elements in 2012, 4 elements in 2013, 3 elements in 2014, 5 elements in 2015, 4 elements in 2016, 6 elements in 2017, 7 elements in 2018, 5 elements in 2019, 3 elements in 2020, 4 elements in 2021, and 5 elements in 2022. Therefore, we can safely conclude that in the past few years, a decreasing trend in the effort to record new elements into the List of Intangible Cultural Heritage in Need of Urgent Safeguarding has been observed.

The Commission, in 2013, summarized the minimum technical requirements (in its decision 7.COM 11) as recorded in the report of the expert meeting on the criteria for inscription on the ICH list [7], but it has not been clarified from UNESCO what process each country must follow to accept national applications for inscription, but neither has it specified precisely what data and metadata should be included in their applications and how they must be stored.

Therefore, the main purpose of the present paper is (i) to record the need for a well-defined process by UNESCO, which must be followed worldwide, (ii) to compare some of the metadata models that have been presented in the past few years in order to find the common data that must be, at least, recorded, and (iii) to specify why blockchain technologies (BT) are better (from a perspective of data security) for storing global ICH data and to present a decentralized application (dApp) which must be used by UNESCO (a

multi-layer architecture of this dApp is going to be presented which has been deployed using a systemic approach).

More specifically, in Chapter 2, two different problems that have been identified by the literature review will be presented. The first problem is that different kinds of data and metadata standards/frameworks have been used to record and visualize ICH data, and this can cause confusion in how to maintain them in the future. To set an example, the literature review is referring to video recording or movement XML for encoding the way that dancers are moving in a folk dance. In combination with music XML for describing and encoding the music/song, it can lead to three levels of a conceptual entity–relationship metadata model, well known as FRBR schema [8].

The second problem refers to storing them, usually, on local servers and providing open access to all others (metadata libraries use these data or these data are linked to the information stored in other digital libraries). However, in the case of the war in Ukraine, there are already 5000+ sites offline, which contained over 50 TB of data of Ukrainian cultural institutions [9] and 246 cultural sites (107 religious sites, 20 museums, 88 buildings of historical and/or artistic interest, 19 monuments, 12 libraries) that had been partially or completely destroyed until March 8, 2023 [10].

This study continues in Chapter 3 with the presentation of the procedure followed in Greece in scripting a new element in the National Inventory (NI) of ICH of Greece in order to emphasize the need to create a standardized procedure to be followed worldwide. Three findings emerged from the study of the procedure and the files of the already inscribed elements. The site does not make any reference to any kind of data and metadata standards/frameworks adopting the UNESCO principle. There is also no proposal for a technological upgrade of the data already submitted in the nomination files. From UNESCO's side, there is a directive for periodic reports (every 6 years in the Representative List of the Intangible Cultural Heritage of Humanity and every 4 years in the List of Intangible Cultural Heritage in Need of Urgent Safeguarding) from member states without clarifying their content which should at least include the technological upgrading of the submitted documents. The third finding concerns the way of storing digital data and an example of the Greek ICH element "Rebetiko" is given as a case study.

All the above findings resulted after the study of the inscriptions procedure in the UNESCO lists and national inventories, as well as the study of the relevant literature, attempted to be studied in the light of systems thinking in Chapter 4 as it is a multidimensional phenomenon and can be influenced by many and different factors/variables. Actually, UNESCO is a global network system and all the state members are regarded as subsystems in order to serve a need, the need to spread knowledge beyond borders. However, how can this global network, where its internal essential parts are interconnected and interdependent, continue working without any malfunctions (be a viable system) in a period of turbulent environmental changes, such as those which have been recorded in the last few months due to the war in Ukraine? How must we store the digital data of the UNESCO lists, so as to safeguard them for the next generations?

In Chapter 5, the last question examined, as to which of the existent technologies could be used not only to ensure data integrity, but also its confidentiality and availability 24/7, is taken into consideration. Many ways to store the data have been suggested, such as (i) centralized ledgers in File Transfer Protocol (FTP) servers, (ii) decentralized ledgers—distributed ledger technology (DLT), (iii) distributed databases using blockchain technology (BT) [11], and (iv) Oracle's Blockchain Cloud Platform—Baas [12] (p. 68) or a more prominent one such as Oracle Blockchain Platform—OBP, etc.

The authors have decided in this article to discuss the decentralized databases in contrast with the centralized ledgers which are currently used to save the digital data of ICH [13]. This choice is based on the situation observed at the beginning of the war in Ukraine. More specifically, when the war began, the electricity stopped in several cities in Ukraine. Moreover, with the arrival of the army, several facilities were destroyed. Facilities such as ministries, universities, museums, libraries, etc. Some of those facilities

were hosting FTP servers/government databases which contained sensitive digital data, such as those of Ukraine's intangible and tangible cultural heritage. Therefore, a new and vital requirement was born. The need to back up all these data as soon as possible. Although in the General Conference of UNESCO, in 2015, there was a declaration for extra reinforcement actions for protecting culture in case of armed conflict [14], the Ukrainian Government was not prepared. The project, which was developed rapidly for this purpose, is called "Saving Ukrainian Cultural Heritage Online" [9] and constitutes the initiative of 3 persons: Quinn Dombrowski (Stanford University), Anna E. Kijas (Tufts University), and Sebastian Majstorovic (Austrian Centre for Digital Humanities and Cultural Heritage) and 1500 international volunteers.

However, can humanity entrust the safeguarding of the ICH of Humanity to the sensitivity and/or initiative of some people or should strategic decisions be immediately taken for the prevention, and eventually for the safeguarding of these data? That is why the main purpose of this paper is not only the need for the creation of a decentralized network (an ecosystem) between UNESCO and each country's Ministry of Culture, local cultural institutes, libraries, museums, etc., but also the need for storing the data of ICH in a public ledger (use of BT) where all the above parties (essential parts of the network) will keep a complete copy of all the records.

Finally, in Chapter 6, some conclusions are extracted which will be the starting point for a new standard that must be adopted by UNESCO that will ensure that the data will be recorded in a predetermined method, which is provided for in the blockchain consensus mechanism.

## 2. Data of ICH and Standards of Metadata: Literature Review

As is mentioned above, any element can be added in one of the five domains from the lists from UNESCO's Convention: (1) the Representative List of the Intangible Cultural Heritage of Humanity and (2) the List of Intangible Cultural Heritage in Need of Urgent Safeguarding (Figure 1), but it must be recorded and visualized in such a way so as to meet the criteria that UNESCO has set. Of course, UNESCO not only has already created proper forms (ICH-01, ICH-02, and ICH-03) for these inscriptions, but it has also predicted how the transition from one list to another can be achieved [15].

In each case, no matter how much the authors searched UNESCO's site, they were unable to locate the actual technical specifications that a new element must meet. For example, let us suppose that there is an interest in recording and visualizing a folk dance. The video file, which must be delivered with the application, according to chapter I.7 and I.8 of UNESCO, Basic Texts, 2020 edition [15], must be created according to MPEG-7 standard or according to another standard? Respectively, which metadata standard must be used for this video? Each standard contains different metadata for the video, and thus for the element itself.

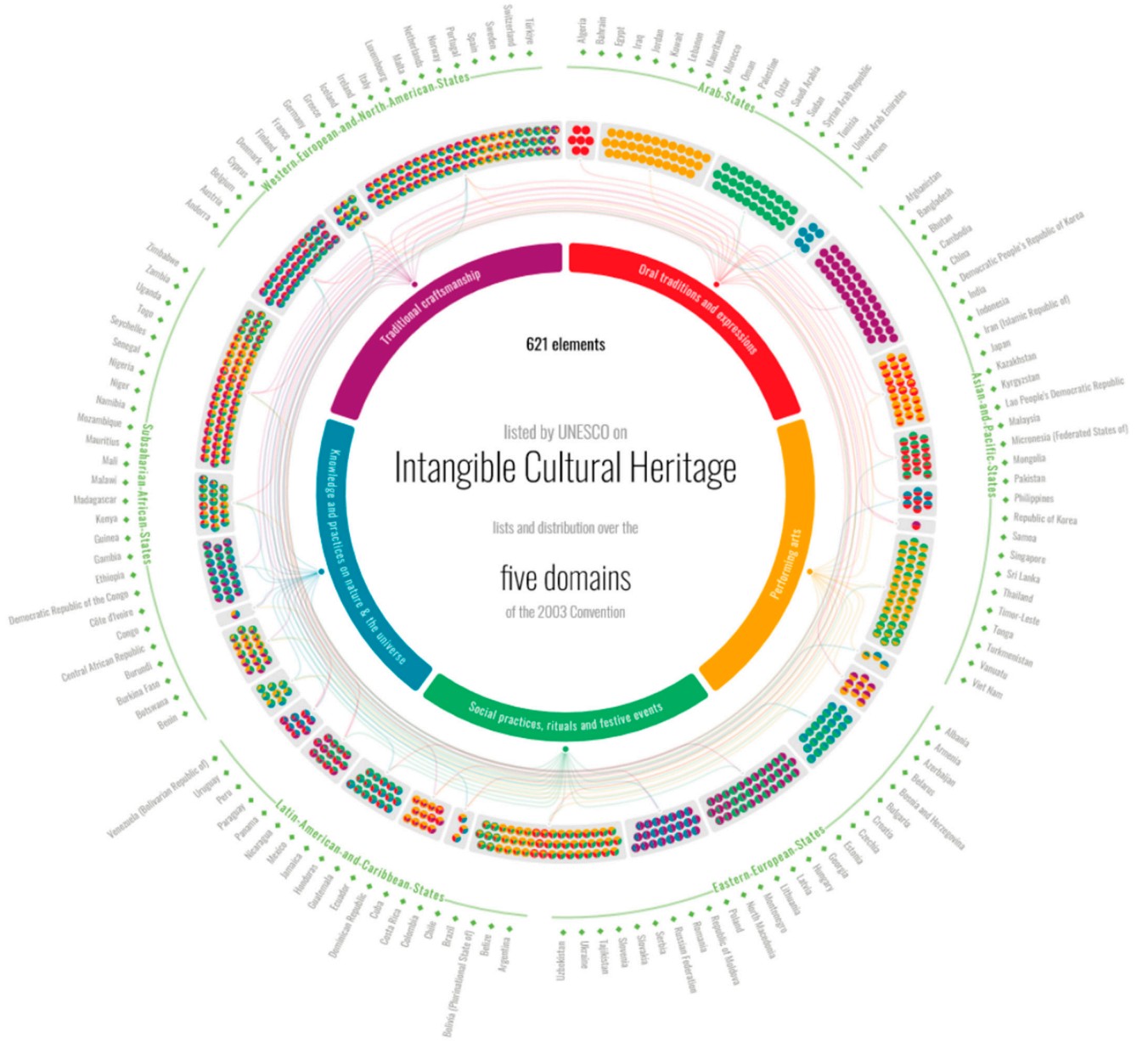

**Figure 1.** Dive into living heritage. The five domains of the 2003 Convention [16]. Note: The number of elements has currently updated to 708 at the end of 2022 [3].

Another finding from the literature review, but also from our own research on UNESCO's site, is that according to chapter 2's operational directives, I.3 § 3-7 of Basic Texts of the 2003 Convention for the safeguarding of intangible cultural heritage [17] (p. 28) is that the possibility of accepting the results/proposals or other material from various projects (related to the recording and visualizing elements of ICH), is recognized.

There have been many projects in the past few years. For instance, the Digital Culture DigiCULT project [18] which was conducted from 6/2020 to 10/2021, in which a new data model was developed for recording a heritage expression. This DigiCULT Data Model (DCDM) not only provides a series of variables/data that must be recorded, as a minimum, in case of applying the inscription of a new element (ICH-Data-Model) [19], but it also explains the file format of the texts, the audio files, the images, the video files, the metadata, and the standards of the metadata that these files must meet (O1-Data-Model) [20]. In Chapter 6.2 of this data model, two different ways to store the data in a database are traced. Either to store in a standard relational database, in which the data and the metadata of

each element are stored in the fields of a table so as to link a heritage element with another heritage element, or to create a relationship between a heritage element and an external entity [21] (p. 133). The second way is to store both the data and metadata of an element in the same FTP server, so as to create a repository of a specific kind of data.

At this point, in order to track the way UNESCO stores the data in its database, we visited the website https://ich.unesco.org/dive/domain (accessed on 18 January 2023) [16] and from the picture in the center of the screen (Figure 1) we chose the domain "Traditional Craftsmanship". Then, we chose the bullet with the name "Tango" which popped up a new window with all the data of this specific ICH element. We traced that the data are stored locally and the video is stored in the public FTP server of YouTube (we must mention that this video is unavailable due to its copyrights), but there is an interconnection between them. In addition, the photos of this element are available by clicking on the "next" button and if we select the hyperlink "See all the information available on this element" we can confirm that the photos are stored locally. We repeated the same process for almost two-thirds of the data (in each one of the five domains) and we were led to exactly the same conclusions about the way that those data are being stored.

In both methods, which have been mentioned earlier, we will face the same problem. The data are stored in a centralized ledger (Figure 2) and due to the open access, which is provided to every user, it automatically becomes very vulnerable to human mistakes or to malicious actions.

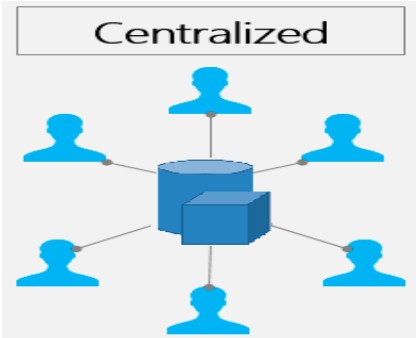

**Figure 2.** Centralized ledger—pro blockchain [22] (modified by the authors).

Other projects are also traced, such as the i-Treasure project [23] which refers to the first above-mentioned method of storing the digital data of ICH and at the same time proposes the development of an open and extendable platform to provide open access and the ability for interconnections with other libraries for research and educational reasons. Similarly, the I-Maestro project, the Cultural Capital Counts project, the U'mista Cultural Centre project [23], and many others are focusing on the first method of storing the data with open access for users, either for educational reasons or for linking them with other data so as to enrich them. In any case, many different ideas have been proposed regarding the way that those data can be analyzed or can be processed with different tools/software so as to enrich the existing ICH elements. The Europeana project introduces a portal through which the user can trace an element from a museum and/or from a library in Europe. However, where are these elements really stored? Each library, museum, and/or local cultural association is responsible for its FTP server which contains all the necessary data and metadata for the cultural elements that it owns. It is therefore clear that each country owns its repositories which are usually placed in museums, libraries, Ministries of Culture, etc.

At this point, each reader can easily understand that as each country is responsible for the data of its own repositories, it simultaneously becomes responsible for (i) defining the procedure for adding any ICH element in its repositories and (ii) for determining the minimum data that should be received from the researcher who has undertaken the task of recording and visualizing this element. These two facts automatically turn a country into an autonomous entity, regardless of the whole or regardless of any problem

that may arise when an element must be interconnected with data from another source (external entity) located in a different country. Those interconnections of the data are often necessary, not only to enrich the data and to document the historical retrospection of this element, but also to prove and to secure its copyrights ownership. As the authors will also mention in Chapter 4, when an element is stored locally without following the same procedure globally, it is mathematically certain that in the future people will run into an interconnectivity problem. More specifically, the interconnection of data from different sources (servers) usually requires special source code (API scripts) as it is very usual that those sources do not use the same programming language.

The necessity of these interconnections has already been traced [24], and so new metadata models must be designed which at the same time must be fully compatible with the most used/well-known standards. One of the most prominent and most used frameworks for metadata is the Dublin Core Application Profile (DCAP) [25] which provides interoperability between the metadata of cultural elements. Of course, there are also many other well-known models, such as the Europeana Data Model—EDM [26] which aggregates the metadata from over 3000 institutions in Europe [27], RLSP Metadata model, CDWA metadata model for arts/architectures/cultural works, CIDOC/CRM for exchanging data/information between cultural heritage institutions all over world, etc. Thus, at this point, we would like to touch upon the diversity of all these metadata schemes.

After comparing the five below items with the metadata, it is noticed that all the metadata models which have been proposed by their authors have some common characteristics.

- Item 1 "Core metadata of intangible cultural heritage digital resources" from the article "Research on knowledge Organization of Intangible Cultural Heritage Based on Metadata" [24]. This item depicts an ICH archive metadata schema containing 23 Dublin Core elements and extended elements, and unifies the information format and mutual mapping relationship of ICH digital achievements.
- Item 2 "Semantic relationships between the CHDE and existing schema classes" from the article "Metadata Model for Organizing Digital Archives of Tangible and Intangible Cultural Heritage and Linking Cultural Heritage Information in Digital Space" [27]. This item depicts a semantic mapping between cultural heritage in digital environment (CHDE) classes and existing related schema classes.
- Item 1 "CRR vocabulary relationships" from the article "Integrated classification schemas to interlink cultural heritage collections over the web using LOD technologies" [28]. Marcondes tested and updated the culturally relevant relationships (CRR) vocabulary and intended to reuse relationships of other vocabularies as the Dublin Core.
- Item 3 "The set of Intangible Cultural Heritage Metadata Standards" from the article "The Metadata standards of Chinese Intangible Cultural Heritages" [29]. The metadata standards for Chinese ICH are a brilliant example because they unify the digital information and mapping relationship. They contain 14 standard metadata names, 67 elements and expanding elements, and 67 field names.
- Item 1 "The proposed metadata for the metadata scheme" from the article "Metadata for Intangible Cultural Heritage—The Case of Folk Dances" [8]. The proposed metadata for a digital choreographic metadata model defines a new metadata standard, based on the Dublin Core, which encodes all the metadata elements referred to in this item.

Of course, these characteristics are nothing more than the minimum data that a researcher of an ICH element must record. For example, (1) the title of the element, (2) the domain that must be registered to (one of the five available domains), (3) the country/region of origin, which is also known as the owner of the copyrights, (4) the type of element (whether it is a folk dance, a common practice, etc.), which we sometimes call "event type," (5) the agent/creator, which in fact is the researchers' personal data, (6) the parts of an element, which are also known as "resources" (the attached files such as videos, pictures, texts, etc.), (7–8) the format of those files and a unique identifier for each file, and

(9) any links with data from other ICH elements or from external entities, and many others. Although those characteristics will be discussed further in Chapter 5, the reader of this paper must be aware so as to understand the necessity for creating a metadata model which will contain specific data fields depending on the type of ICH element which is going to be inscribed.

### 3. The Implementation of 2003 Convention in Greece and the National Inventory of ICH of Greece

How is the 2003 Convention implemented by each State Party? The implementation of the Convention is determined by an institutional and a legal framework. The Convention draws up national inventories where the heritage of each member state is recorded. Subsequently, some of the native cultural expressions from the national inventories are qualified under standardized procedures and registered in the international lists.

According to Article 12 of the Convention, "To ensure identification with a view to safeguarding, each State Party shall draw up, in a manner geared to its own situation, one or more inventories of the ICH present in its territory" [2]. The registration of an element in the national inventory/-ies is one of many and varied actions that can contribute to the study, preservation, and promotion of ICH.

The aim pursued through the NI [30] is to give the bearers of ICH, such as communities, groups, and even individuals, the opportunity to share their own cultural experience, to talk about their collective identity, and to discuss ways of studying, promoting, and safeguarding their ICH on a national and international level. However, on the basis of which criteria will an ICH element be recommended for registration in the NI? The most important fact is that the element is proposed by the bearers themselves because it is recognized as an important element of their collective identity and memory, as it is integrated into their living cultural experience. This cultural element transmitted from one generation to another contributes as a connecting bond to the continuity and cohesion of communities as a whole. When an element of ICH is lost, the road that connects us with the past is lost, but also an open window to the future is shut, especially for the younger generations.

However, the methodologies adopted to create these national inventories vary from one country to another, because the Convention is flexible in the way each member country promotes different aspects of its ICH, as well as the policies it follows to safeguard and record them. In Bulgaria, questionnaires are distributed to local communities to gather information, while in Japan and Brazil, ethnographic research is preferred. In Greece, ethnographic research and interviews with the ICH communities' bearers are mainly preferred. Perhaps, the most innovative method of gathering material is recorded in Scotland and consists of an online archive of photographic and audio-visual documentation. It is open to the public in the form of a wiki and collected via participatory approaches [23].

Greece, which ratified the Convention in 2006, has drawn up the "National Inventory of Intangible Cultural Heritage of Greece", which is the filing of elements of ICH in the form of an inventory and provides an up-to-date presentation of the ICH of Greece. The National Inventory (NI) includes 102 elements so far (Figure 3, December 2022). In addition to the NI, scientific and other respective bodies have already created or will surely create other inventories for Greek ICH in the future. However, when one refers to the implementation of the Convention in Greece, they refer only to the NI.

Each element proposed for registration in the NI of ICH of Greece must comply with the general principles of both the 2003 Convention, as dictated by the promotion of peaceful coexistence between people, the sustainable development, respect for the rights of living organisms, etc., and the NI of ICH itself. The information related to each element of ICH inscribed on the NI is depicted in the "form of the element of ICH of Greece", which can be found in the following link: https://ayla.culture.gr/odigies-syggrafis-deltiou-apk/ (accessed on 18 January 2023) [31]. Each form is divided into eleven fields, each of which gives the opportunity for particular aspects of the element to develop. Several of these

fields include multiple subfields. The eleven fields are (1) brief presentation of the element, (2) identity of the element's bearers, (3) detailed description, (4) space and equipment related to its performance, (5) products or general material objects resulting from its performance, (6) historical evidence, (7) current significance of the element, (8) safeguarding and promotion, according to basic bibliography, (9) contact details of the form's authors, (10) additional documents, contact details of the form's authors and date of submission, and (11) date of the last updated form.

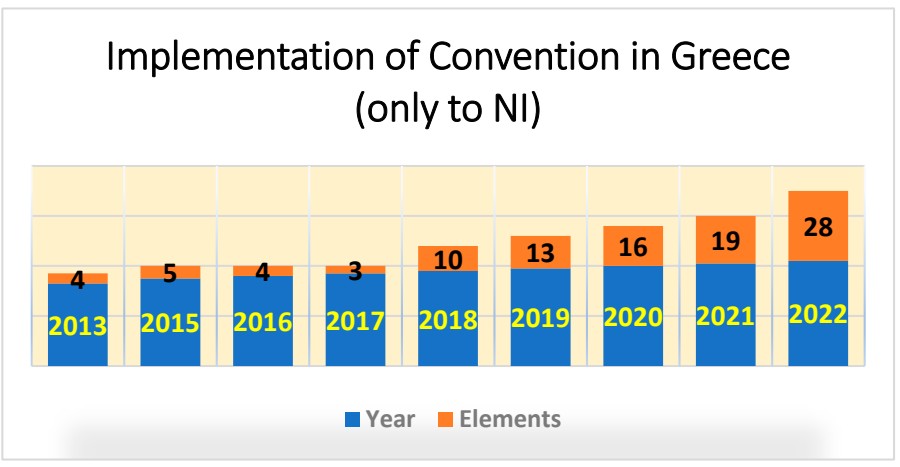

**Figure 3.** Diagram of the inscribed elements in the National Inventory of ICH of Greece (2013–2022).

The NI is not structured based on any specific criteria, such as thematic or geographic. Visitors can browse by choosing a topic of interest and create their own personalized browsing path. During this tour, one element can direct them to another element (external data of information such as texts, photos, videos, etc.) or to dive deeper by consulting the relevant literature or even to contact the bearers of the specifically referred ICH community.

The Inventory is completed and updated on an annual basis following a standardized procedure following an initiative by the community of the element's bearers. Greece has been a state member of the 2003 Convention since 2007 and has to follow the rules of this Convention. The main body for implementing the Convention in Greece is the Directorate of Modern Cultural Assets and Intangible Cultural Heritage (MCA&ICH) of the Ministry of Culture and Sports. Additionally, in 2012, a National Scientific Committee for the Implementation of the Convention was also established. The Law for the Protection of Antiquities and Cultural Heritage in General (2002) provides an overall framework for the safeguarding and management of heritage, including living heritage.

As mentioned earlier, the enrichment of the NI of ICH of Greece is an annual process, open to all and structured in three stages [32] as depicted in Figure 4 (the below system dynamics model is presented without the mathematics and its complete form, because it is part of the PhD thesis of the second author and as such cannot be published completed in this article):

Stage A: Includes the "Statement of Intention to Submit ICH Elements" and examines whether the proposed element for registration meets the basic inclusion criteria set forth on the one hand by the 2003 Convention and on the other hand by its implementation policies in Greece. If the interested element is not yet "ripe", it may not continue on its own initiative in subsequent stages. The statement is sent by email and indissolubly linked to the email address ayla@culture.gr.

Stage B: Is called "Publication of Statement of Intention to Submit ICH Elements". The statements of Stage A that do not comply with the principles of the UNESCO 2003 Convention and its implementation policies in Greece are rejected and receive a response by email (B.1. Rejection). Those that meet them are posted on the ayla.culture.gr, the ICH website in Greece (B2: Posting), in order to inform their authors and the public that they qualify for Stage C. Any interested bearer of the element may contact the authors of the

statement to contribute to the enrichment of the element (B.3. Public consultation). It is pointed out that the acceptance of Stage B does not imply the registration of the element in the NI of ICH of Greece.

Stage C: Is titled as "Completion and Submission of ICH Elements to MCA&ICH". As long as the community of bearers has completed the consultation period required, it must send the completed form to MCA&ICH within a specified period of time following the detailed instructions given. The examination of late submitted statements is postponed until the following year. After the submission, the certified employees of the MCA&ICH collaborate with the authors to improve it, if it is necessary. When the statement is considered to be meeting the specifications governing the National Inventory, then the statement and the other accompanying documentary material (such as photographs, video, etc.) are submitted to the C.1. National Scientific Committee for ICH. The Committee either reports negatively (C.1.1. Rejection) or positively (C.1.2. Registration by Ministerial Decision in the NI of ICH of Greece). In any case, the authors of the statement are informed in writing.

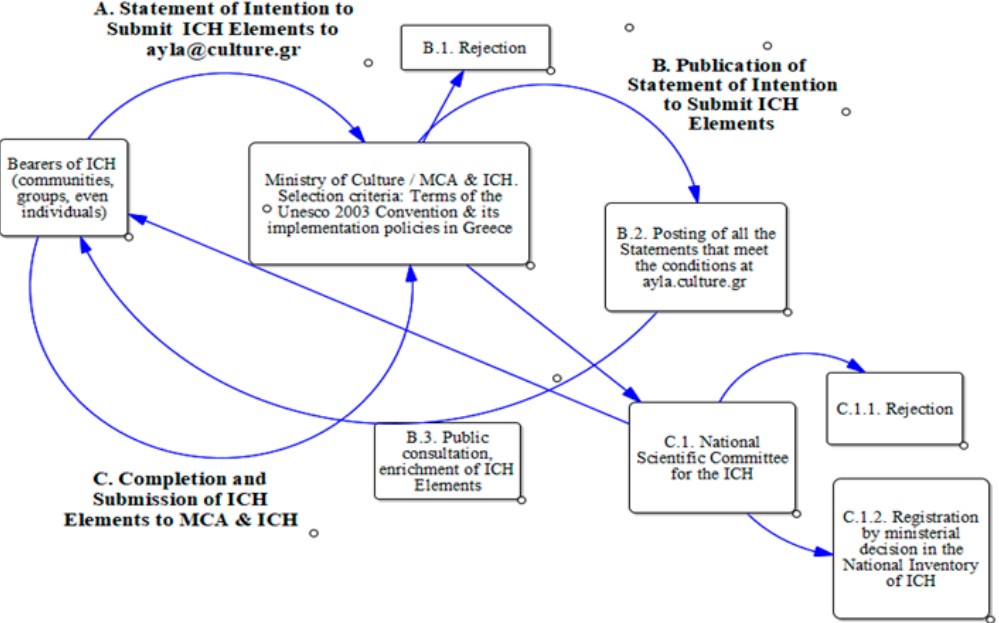

**Figure 4.** Mapping of the inscription procedure of ICH elements on the National Inventory of ICH of Greece.

The inscription of an element on the NI of Greece is by no means an end in itself. The inscription aims to highlight the importance of ICH and to present it to the members of the community as a connecting link to their identity, but also to motivate them to act for its promotion, as well as to the wider public, domestic and international.

The next step is the submission of a nomination file of the element already registered on the NI of Greece to one of UNESCO's lists from the 2003 Convention, which is part of a wider planning that takes into account various parameters. In addition, since these are international convention lists, the certified services of the Ministry of Foreign Affairs are also involved.

Greece has so far inscribed nine elements of ICH on the Representative List of the Intangible Cultural Heritage of Humanity:

- Mediterranean Diet (joint nomination file: Cyprus, Croatia, Spain, Greece, Italy, Morocco, and Portugal), 2013 (UNESCO 8.COM 8.10) [33];
- Know-how of Cultivating Mastic on the island of Chios, 2014 (UNESCO 9.COM 10.18) [34];
- Tinian Marble Craftmanship, 2015 (UNESCO 10.COM 10.b.17) [35];
- Momoeria a New Year's Celebration, 2016 (UNESCO 11.COM 10.b.16) [36];

- Rebetiko, 2017 (UNESCO 12.COM 11.b.11) [37];
- Art of dry stone walling, knowledge, and techniques (joint nomination file: Croatia, Cyprus, France, Greece, Italy, Slovenia, Spain, and Switzerland), 2018 (UNESCO 13.COM 10.b.10) [38];
- Byzantine chant, 2019 (UNESCO 14.COM 10.b.2) [39];
- Transhumance, the seasonal droving of livestock along migratory routes in the Mediterranean and in the Alps (joint nomination file: Austria, Greece, and Italy), 2019 (UNESCO 14.COM 10.b.2) [40];
- August 15th (Dekapentavgoustos) festivities in two Highland Communities of Northern Greece: Tranos Choros (Grand Dance) in Vlasti and Syrrako Festival, 2022 (UNESCO 17.COM 7.b.11) [41].

Additionally, Greece has also inscribed one element in the Register of Good Safeguarding Practices:

- Polyphonic Song of Epirus, 2020 (UNESCO 15.COM 8.c.4, 2020) [42]

Studying the files of the Greek-inscribed elements on the NI, different data and metadata are found from one element to another, and so three findings emerged: the first is that even if there are many proposals in the bibliography, as mentioned in Chapter 2 above, regarding what data and metadata should be recorded in each type of ICH element, no proposal has been defined by the Greek side following the UNESCO policy. For example, referring to a folk dance, in one case, the MPEG-7 standard was used for storing a video [8] and in another case the DCT-based standard was used for the same purpose. This difference lies in the fact that UNESCO has not defined: (1) the format of the video that must be saved in each case (DV or 1080i HDV or AVCHD or XDCAM HD, etc.) and (2) the minimum data fields of the elements that must be stored in each case (title, creator, source, ownership rights, subject, etc.).

The second finding is that although documentation and recording technologies are rapidly improving over time, there is no proposal in the Greek literature for the technological upgrading of the documentation that has already been submitted in the oldest registered cases to the NI. To be more accurate, even if the Convention suggests in Article 29 that State Parties have the obligation toward the Committee to submit reports on the legislative and other measures taken for the safeguarding of ICH in their territories every five years, in the first periodic report from Greece, in 2015, it is generally stated that "Each entry should be updated every five years through an open call to the bearer community". There is no reference to any sort of technological upgrade of the documentation. The element with the title "Polyphonic Song of Epirus" [42], inscribed, in 2020, in the Register of Good Safeguarding Practices, is set as an example. Cassettes, and discs of Bakelite and vinyl since 1928 serve as proof of its documentation. It is not mentioned anywhere that the audio material, beyond the citation of the historical and archival documents, has been transcribed into audio of contemporary technology.

The third finding is the way that digital data are stored. When an item is registered from national inventories to UNESCO lists, its storage is usually on local servers containing the digital cultural heritage of the origin country and open access is provided to all others.

In order to trace the digital metadata path of an inscription from the national inventories to the international UNESCO lists, the procedure of inscription of an ICH element called "Rebetiko" (a kind of Greek music and song) from the NI of ICH of Greece (2016) to the Representative List of the Intangible Cultural Heritage of Humanity (2017) is presented as a case study in three stages:

1. The researcher collects all the necessary data (photographs from the dancers and their facial grimaces, video recordings, historical records of this kind of music, recordings about the melody and the words of the songs, interviews of the participants/dancers/musicians, etc.) and submits its application to the department of MCA&ICH of the Greek Ministry of Culture and Sports.

2. If the application meets the minimum requirements, as they have been set from the NI and UNESCO, then and only then, can this element be inscribed in the NI of ICH of Greece (in fact, it will be stored to a local file server).

3. The Greek Ministry of Culture and Sports examines whether this element meets the criteria to be inscribed in UNESCO lists. If so, then it will be passed to the competent department of the Greek Ministry of Foreign Affairs, which in turn will forward the application to UNESCO for further examination so as to be set as an element of the ICH of Humanity. Then, UNESCO will create a new record in which some of the metadata will be saved into its file server (such as the title of the element, an abstract, the creator, the subject, the source, etc.) and, regarding the files of videos or photos, it will store the necessary URI so as to redirect the reader to the proper local file server (where those data are stored). However, if each local server, where the digital data of ICH are being stored, is examined as a separate system, many interdependency relationships would appear among them due to those redirections.

To sum up, UNESCO is a global network (such as a hyper system) where all those local file servers (subsystems) are interconnected by relationships of interdependence (communication channels) in order to serve a need, the need to spread the knowledge beyond borders. However, how can this global hyper system, where its internal essential parts (FTP servers—subsystems) are interconnected and interdependent, continue working without any malfunctions (be a viable system) in a period of turbulent environmental changes, such as those which have been recorded in the past few months due to the war in Ukraine? How must we store the digital data of the Representative List of the ICH of Humanity, so as to safeguard them for the next generations? Indeed, it is a multidimensional phenomenon that will be approached under the light of systems thinking.

## 4. A Systems Thinking Approach

According to the general system theory of Ludwig von Bertalanffy (1934), "A system is an entity which maintains its existence through the mutual interaction of its parts" [43]. A system consists of three parts: subsystems, subsystems interconnection, and purpose of the system. That is why UNESCO could be considered a system and each country that applies for a new element to be registered into one of UNESCO's lists must be considered an essential part of this system and regarded as a subsystem. The purpose of the system is to safeguard and preserve the worldwide ICH, which will be succeeded through the mutual interconnection of the subsystems.

As was mentioned in Chapter 2, each country is responsible for entering the data of its ICH elements into its repositories. However, sometimes, there are elements that must be interconnected with data from other repositories which may be situated in different countries (subsystems interconnection). The fact that each country is entering the data of its ICH using different data and metadata schemes is going to lead to a huge problem of interoperability.

Moreover, another fact, which will also cause a problem, is that when two different countries are going to inscribe a new element in UNESCO's lists and supposing that these elements belong in the same domain (for example, they are folk dances), then the fact that those two countries are registering different metadata for an element of the same domain may create confusion for future readers. In each case, the source of the problem is that those countries are acting as separate entities/systems, even though when they apply to UNESCO, they automatically become parts/subsystems of the same broader entity.

As Russell Ackoff stated, in 1981, "System is a single set, which has one or more defining functions and consists of two or more essential parts, fulfilling three basic rules" [44]. In the prementioned case, the essential parts are the two countries which intend to inscribe a new element in UNESCO's lists. The first and most critical "defining function" is the recording and the visualization of their ICH, thus the second one is the registration of its ICH into one UNESCO list. Knowing that the three basic rules are:

1.  Every essential part/subsystem of the system can influence the behavior and/or the properties of the system;
2.  None of the essential parts/subsystems can have an independent effect on the basic one or more function(s) of the system;
3.  When the individual parts of the system are organized into subsystems, then they have the same properties as the essential parts.

It is profound that the sustainability of these elements over time, regardless of the environmental changes (such as the war in Ukraine, for example) depends entirely on the proper behavior of the subsystems.

The cybernetic framework of the viable systems model (VSM) of Stafford Beer is also used to evaluate the traditional hierarchical edifice of the organization of UNESCO, identifying the functional incongruities and communication disconnects, which hinder synergies and the organization's overall resilience [45]. According to the VSM, which is described as a "holistic model involving the intricate interactions of five identifiable but not separate subsystems" [46], a system depends on the harmonious operation of these five identifiable and individual subsystems: (1) primary activities, (2) conflict resolution, stability, (3) internal regulation, optimization, synergy, (4) adaptation, forward planning, strategy, and (5) policy, ultimate authority, identity. By the term "harmonious operation", we define the operation of subsystems according to some predetermined conditions. When, for example, a country (subsystem 1) adds a new element of ICH (element O) to its repositories, that means that the researcher who was responsible for the recording and the visualization of this element has followed a predetermined scheme/data and metadata model. This implies that the rest of the countries that are intending to make a relevant entry of an element of the same domain to their repositories must also follow exactly the same data and metadata model/scheme. Those countries are parts of subsystem 1 of the element (O) of the VSM and their "defining functions" are the 1a, 1b, etc. of the element (O—operations). Since UNESCO has to determine the data and metadata model which must be used in each case and to determine the forms that must be completed and submitted for the registration of a new element in its lists, that means that UNESCO is the subsystems 2, 3, 4, and 5 of the elements (M—management) of the VSM. Every user of or visitor to UNESCO's repositories who interacts with the material of these repositories belongs to the environment (E) of the UNESCO system and, all together, they are creating an ecosystem (Figure 5).

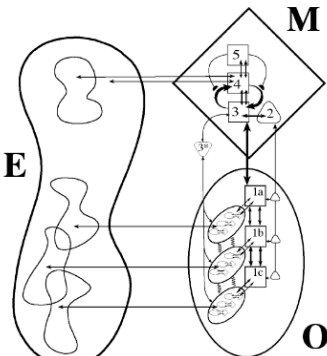

**Figure 5.** The diagram shows the three main elements of each VSM: the operation (O), the management (M), and the environment (E) [46].

We can also consider the members of subsystem 1 as parts of the environment (E), only when they act as users of or visitors to UNESCO's repository. In order to give a clear picture of the subject matter, here is an example. Let us suppose that a new element is going to be stored in UNESCO's repository. The country which owns the copyrights of this element participates as subsystem 1 of the VSM. While this element is under approval, any employee of the service that applied for this element (for example, an employee of

the Ministry of Culture) can visit UNESCO's repository (website) as a simple user whose purpose is to find out whether the process of approval has been completed and at the same time to identify whether there are any errors in the element's data. In this case, the employee acts as a simple member of the environment (E). However, if he discovers any error and then consequently applies for any corrections, he does not act as a member of the environment (E), but as a member of subsystem 1.

As Stafford Beer also mentioned, the sustainability of a system is based on the ability of this system to deal quickly with the changes in its environment by making improvement interventions to learn from those changes and of course to keep evolving within it. Therefore, even if one or more countries become involved in a war (such as the example of Ukraine), which can lead to turbulent changes in the environment (E) of this system, the data of ICH cannot be lost provided that the BT would make many copies of the data which will be kept on the servers of all the countries/members worldwide (parts of subsystem 1).

Given that the essential parts (member states) of a system (UNESCO) behave in a non-predetermined way, according to system thinking, the number of alternative outcomes from the interdependences of these essential parts could be an exponential number. On the contrary, this number is noticeably reduced and not only can it be studied, but it may also (in some cases) be predicted, especially when the behavior of the essential parts is predetermined. This fact emerges from the reduction in the dynamic complexity. Thus, in order to reduce this complexity, UNESCO ought to determine a better and more specified way of recording and visualizing the ICH data, as well as to determine the use of BT to store these data as is graphically depicted in the next figure (Figure 6).

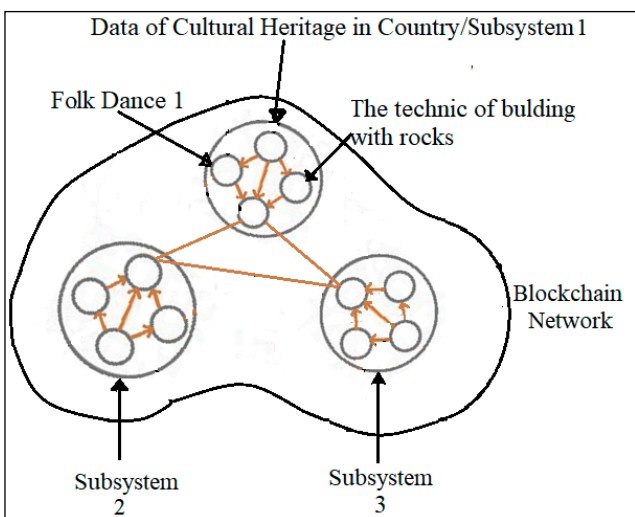

**Figure 6.** Blockchain network [47] (modified by the authors).

At this point, the readers of this article may have questions which remain unanswered, such as can this technology be adopted by all member countries of this ecosystem? and if so, what difficulties may arise in the procedure of the adoption? These questions, as well as the description of the preparatory actions that UNESCO must undertake, will be analyzed in the next Chapter.

## 5. How and Why Blockchain Technologies Must Be Used for Storing the Data of the Intangible Cultural Heritage of Humanity

In the previous chapters, we have mentioned that when a country is called upon to deal with sudden and turbulent changes in its environment, it usually struggles. A living paradigm is that of Ukraine where 50TB of digital data of local ICH were in danger of becoming lost due to the war. The fact that the data were not lost is not because of the preparation of the country to deal with such a threat, but is clearly due to human intervention following the initiative of specific people [9]. Ukraine, in the past, had applied:

- In 2013, for the registration of the "Petrykivka decorative painting as a phenomenon of the Ukrainian ornamental folk art" (UNESCO 8.COM 8.29) [48] on the Representative List of the Intangible Cultural Heritage of Humanity;
- In 2016, for the registration of the "Cossack's songs of Dnipropetrovsk Region" (UNESCO 11.COM 10.a.5) [49] on the List of Intangible Cultural Heritage in Need of Urgent Safeguarding;
- In 2019, for the registration of the "Tradition of Kosiv painted ceramics" (UNESCO 14.COM 10.b.40) [50] on the Representative List of the Intangible Cultural Heritage of Humanity;
- In 2021, for the registration of the "Ornek, a Crimean Tatar ornament and knowledge about it" (UNESCO 16.COM 8.b.45) [51] on the Representative List of the Intangible Cultural Heritage of Humanity;
- In 2022, for the registration of the "Culture of Ukrainian borscht cooking" (UNESCO 5.EXT.COM) [52] on the List of Intangible Cultural Heritage in Need of Urgent Safeguarding.

Apparently, these five elements cannot occupy a capacity of 50TB. Therefore, although Ukraine had stored the information of this capacity (50TB) in its repositories, nevertheless only the data of these five elements met UNESCO's criteria and were stored in multiple repositories, thus avoiding the risk of being lost.

Let us dwell a little more on the expression "were stored in multiple repositories". We therefore understand that if we store global ICH in more than one copy by using, for example, the BT which is storing all the data in a public ledger and the mining nodes of the ecosystem keep a full copy of the data, this will act as a countermeasure to a possible threat, such as that of war. If we take it a step further, it should be defined that when we store the data of each element and we also keep any information related to the owner of its copyrights, then each miner, node of this ecosystem could keep a complete copy of all the elements (even of those which do not own their copyrights). However, in the long run, we will have to learn how to manage and maintain the availability of the data without losing, at the same time, their integrity or their confidentiality (also known as the CIA of data). However, how possible and how difficult is the creation of a new network between UNESCO and the local bearers, such as libraries, archives, museums, or cultural associations?

In order to proceed with the creation of a new decentralized application (dApp), there are certain criteria that must be met. The topology of the network should therefore be initially determined based on the nature of the users of this network. Therefore, it should be decided which entities are going to participate in this network. As we mentioned earlier, if we want to save the ICH of Humanity, we must include in this network all the countries, regardless of their language, their religion, or other limitations. In each country/member of this network, there will be some participants who are directly related to the culture such as, for example, the museums, the libraries, the local cultural associations, history teachers/professors, archaeologists, etc. These participants in BT tend to be called miners and mining nodes [1] or, depending on the consensus mechanism, they can be called alternate names.

There is also another category of participants, those who are just interested in history and culture, purely for educational reasons, such as the tourists/visitors to a country and they will be called nodes or economic nodes of the network. It is therefore understandable that they should be separated by using, as the main criterion, the reason for their visit to UNESCO's repositories of ICH. Therefore, the decentralized topology (Figure 7—DLT) should be used instead of the distributed one, since this topology allows us to separate users' rights (read only or read/write). Consequently, the users with the most rights will be those who apply for the registration of new elements in UNESCO's repository of ICH.

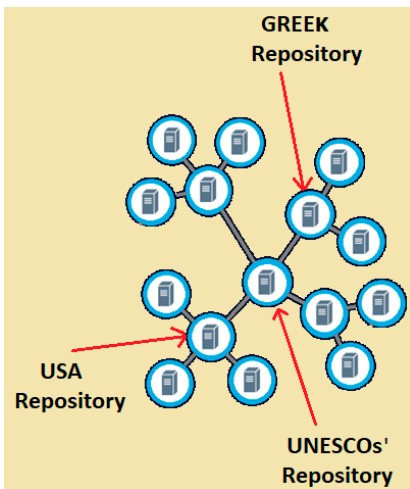

**Figure 7.** Decentralized topology [53] (modified by the authors).

Having determined the topology of the network and the type of users, we must also determine the type of the consensus mechanism of this dApp. In order to determine it, we need to proceed in a preparatory step. We must identify and record the services that this dApp offers in order to decide whether we can use one of the already implemented consensus mechanisms or if we should develop a new one. As we mentioned earlier, the main purpose of this application will be the storage of all the elements of ICH, regardless of their country of origin. With this service in mind, we traced two well-known consensus mechanisms that respond to the need of the dApp that we are proposing to be developed.

The first one is the proof of space/capacity (PoC). When the Ministry of Culture of a country requests the registration of a new element (we are going to call this entity a prover (P)), UNESCO's employees, who are responsible for checking the completeness/integrity of the data of this element (those entities are going to be called verifiers (V)), will proceed with the verification of the stored data on the copy of the prover (data F of size N, where N represents the whole copy of the data and $F \leq N$) and if they pass the verification procedure, then and only then, will the prover (P) be able to proceed with the registration of this element in the next block of the blockchain.

The second consensus mechanism, which can be alternatively used, is the proof of elapsed time (PoET), as it is closer to the "principle of a fair lottery system". That means that all the miners will have equal chances of being selected as the next validator/adder of the block, but is also better versus the PoC, as it needs less electric power and, these days, the use of green technologies is imperative. In addition, the fact that there are two different kinds of consensus mechanisms that can be used in this dApp compels us to choose the Hyperledger Fabric, a flexible dApp development and testing platform.

Since the main purpose of this dApp is to store the data of the ICH of Humanity, it is vital at this point to specify exactly what data will be stored in the blockchain. Our ultimate goal is not only to calculate the size of these data, so as to identify whether we can store them "on the chain" or if we have to use some other storage technique (off chain storage), but also to define the fields of data that must be stored in each type of element. As we discussed previously, there are many different types of elements. For each of these types, we will have to store specific information. For example, in the recording and visualization of a folk dance [8], we will have to store a video file in which the complete movement of the dancers is recorded. The size of this file depends on the video format and on the duration of the folk dance. The most common video formats which have been used in the last few years are MKV, MP4, MOV, H.264, WMV, AVI, AVCHD, etc., and an average duration of a folk dance can be up to 3 min. Therefore, the combination of at least those two variables leads to the creation of a video file where we can calculate the size of this file by multiplying (1) the bit rate, (2) the duration, and (3) the compression ratio, where

the bit rate equals with the multiplication of (1) the frame size and (2) the frame rate [54]. Thus, an MP4 video file with 720p (HD) resolution (bit rate = 1 megabyte per second) and a duration of about 3 min will be approximately 15 Mbytes.

Of course, in order to have a complete record of this folk dance, we must also store some historical data. These data can be stored in either a DOC/DOCX file, which is usually up to 0.5 Mbytes, or in a PDF file, which is usually 1.6 Mbytes [55]. If we add this data size to the previous one and at the same time challenge our reasoning behind it, we will find out that there is also a need to store a set of photos (1) of the dancers' traditional costumes, (2) of the dancers' figures, and (3) of the expressions on their faces as it is important to depict their physical and mental condition while they are dancing. Those image files can be stored in JPEG, PNG, or TIFF format and their size can be calculated by multiplying (1) the pixel count and (2) the bit depth, where the pixel count equals with the multiplication of the width of the image and the height of this image [56]. Usually, the size of an image can be between 0.5 Kbytes and 2 Mbytes.

It can easily be inferred that if we add all the above data sizes as well as (1) the size for the melody of the folk dance, (2) the size of the file with the song lyrics, and (3) the size of the file with the interviews (that the researcher may have taken from the audience), then the final size of this file is going to be over 20 Mbytes. Now, let us imagine how many Greek folk dances there are and try to multiply them by the size of 20 MBytes. Moreover, the folk dances which belong to the domain "performing arts" are not the only ones. There are many other kinds of dances in this domain and imagine how many other elements exist in the other four domains which have been presented in Figure 1.

Therefore, if we try to store all these data "on chain" how could such an act be achieved? Let us suppose that each element (with all its attached files) consists of a unique transaction (each transaction will be symbolized as "Txi", where I = 1, 2, 3, etc.). For instance, the transaction which has been created from a certified employee of the Ministry of Culture of a member country (a unique subsystem of the ecosystem) is going to be the "Tx1". If the data have not been maliciously modified, then the signature check will be completed successfully and afterward, it will be combined with the hash value of another element (for example, Tx2) in order to create a Merkle tree [57], such as the one shown in Figure 8. This Merkle tree consists of the data of the elements which will be saved in a specific block. Thus, if we take into consideration the information which was analyzed in the previous paragraph regarding the size of each element, we understand that after a short time, the size of the blockchain will be very difficult to be managed. In other words, if we try to search a specific element in order to read its data, the server's response time will take too long.

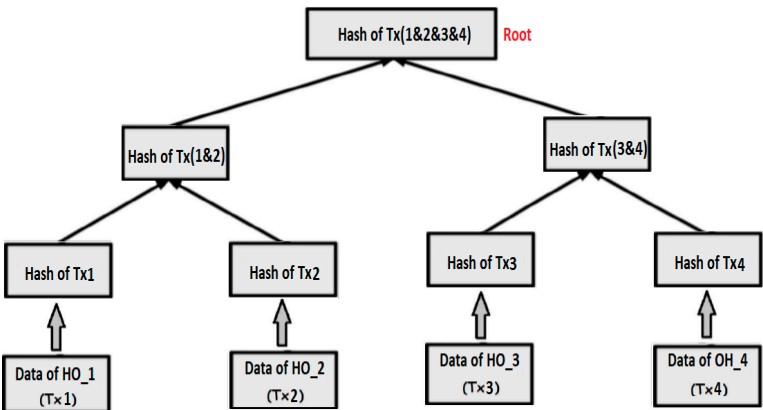

**Figure 8.** Merkle tree [57] (modified by the authors).

On the other hand, if we store all the data of a Txi (video, images, texts, etc.) in the server of the Ministry of Culture of the country which owns the copyrights of this element and we only push the hash value/hash sum of this Txi on the next block of the repository

of the ICH of Humanity (the public ledger), then we will have managed to reduce the size of the stored data on the blockchain, as each hash sum contains only the 64 digits of the Txi string (32 bytes data) [58]. In this case, we have used an "off chain" storage technique which is vulnerable to threats, such as the one of war, as the data are not located on the chain (they are not stored on the public ledger), but in the local server of the Ministry of Culture of the country which owns the copyrights of the elements (all the miners keep a copy of the hash sum of this element only). The fact that the size of the block is a lot less than in the previous case, makes searching for an element much easier and sufficiently faster. Of course, many techniques can be used in conjunction in order to keep these data safe. For example, there can be a separation of repositories by geographical criteria so as to oblige countries to maintain a complete copy not only of its own data, but also of the data of its neighboring countries.

At this point, we are going to use systems thinking, in order to examine the data that we must store in each case. An element must be categorized in one of the five domains of UNESCO's ICH; thus, if we examine the "Building and use of Expanded dugout boats in the Sooma region" (16.COM 8.a.2) [59] which belongs to the domain "knowledge and practice on nature and the universe", we will discover that different data are stored from those which must be stored for the case of the folk dance. In this element, we do not have to store any DOC file or other kind of files for the interviews of the audience, as there is no audience in this process, it just describes how to build a dugout boat by yourself by using a single tree (usually an aspen tree). Although this element describes something a lot different from the previous one, nevertheless they have some similarities. For example, in the folk dance case, we must store photos showcasing the dancers' costumes, whereas in the case of the boat manufacturing, we must also store some photos but in this case including the tools that are going to be used. In both cases, a video must be stored which records the whole process, but in the case of building a dugout boat, the video duration is going to be much longer since it records a process which might take hours or even days. Thus, a new need arises. The need to define exactly which data fields must be stored in each one of those five domains. In blockchain, the fields of the data and the type of the variables must have been defined from the stage of the design of the Data Layer [12] as those fields are going to be used in the interfaces of the dApp which will be designed in the Application–Presentation Layer (Figure 9).

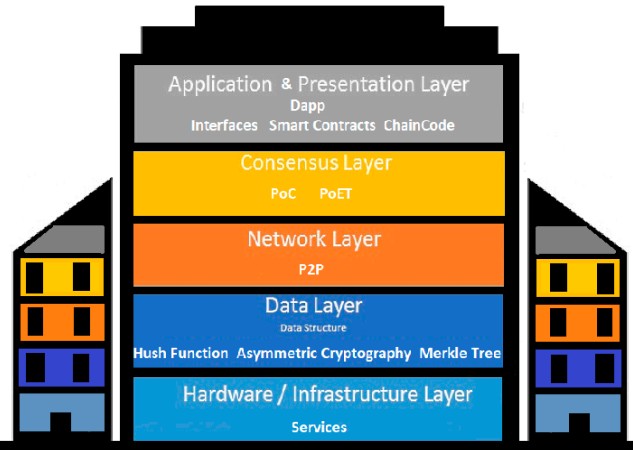

**Figure 9.** DApp layered architecture [12] (modified by the authors).

In Chapter 4, we mentioned that a system, or otherwise an entity, consists of at least two essential parts. Therefore, if we consider the chain of the blocks as a system, automatically those blocks are the subsystems of this system. Correspondingly, each block consists of more than one element. Therefore, in reality, every element is a part of the block's microcosm. This automatically implies that we cannot break an element into independent data (video, texts, photos, etc.), but we must manage it as an entity. That is why we must

use a proper programming technique such as "smart contracts" in order to create five different types of smart contracts, one for each domain of UNESCO's ICH. The advantage of this technique is that we can define various rules/criteria that must be met in each case before storing an element in the public ledger. Of course, for each one of these five kinds of contracts we must define exactly their data fields, the format of the video files, the songs, the images, etc., as well as other fields concerning the copyrights, the data of the research which took over the recording, the visualization of these elements, etc. Due to the fact that the purpose of this paper is only to mobilize the stakeholders/decision makers about how those new technologies can be used for ensuring the ICH of Humanity, that is why we are not going to delve further into the data type issue.

Each element depending on the domain that it should be registered to is going to be converted into a smart contract. This smart contract must meet some predefined criteria (that the stakeholders/decision makers must define) so as to be promoted to the next level. Each block, depending on its size (which is one of the criteria that must also be defined by the stakeholders/decision makers), will contain at least two smart contracts. If those smart contracts refer to the same domain, then if we want, we can package them together into a "chain code" [12] (p. 39).

As you may have already noticed, there are a lot of issues that must be defined appropriately in the design stage of the proposed dApp. Thus, a project team must be established by stakeholders all over the world, which will have a direct relationship with culture and will be the ones competent enough to determine and manage any issue that may arise during the process of developing such an application. After all, the need to have such a "project team" during the process of designing an ecosystem and the deployment of the dApp that is going to serve this ecosystem has already been recorded in the literature [47].

## 6. Conclusions

In the beginning, we talked about the vision of UNESCO and the criteria it has established in order to rescue an element of the ICH of Humanity. We support that the nomination of an element of the ICH of Humanity should be evaluated both on a socio-political economic level and also on a level of content completeness and technical specifications. Studying the files of the elements of the ICH of Humanity, one is struck by how unequal the distribution of elements is in UNESCO's lists during the consideration period (2008–2022). More specifically, the Representative List of the Intangible Cultural Heritage of Humanity includes 599 elements, the List of Intangible Cultural Heritage in Need of Urgent Safeguarding includes 76 elements, and the Register of Good Safeguarding Practices includes only 33 elements [3]. What is more is the fact that in recent years, humanity has faced a series of war conflicts (Ukraine, Iraq, etc.), natural disasters (such as the COVID-19 pandemic, earthquakes, hurricanes), and thus, in our opinion, priority should be given to the List of Intangible Cultural Heritage in Need of Urgent Safeguarding. It is observed that the number of items registered in this list during the years 2008–2022 shows initially a decrease and stability afterward, although it should follow an increasing course. Emphasis should also be given to the Register of Good Safeguarding Practices, in which the average amount ranges between one and five elements per year (very low interest).

However, based on our research, both on its website and in UNESCO's meeting minutes, it was found that there are no clear terms/rules regarding the technical specifications of the data that substantiate an element. It is self-evident that the lack of clear technical specifications creates confusion for stakeholders who apply for the induction of a new element of ICH, but also for their evaluators. Although in the past, several projects have been funded regarding the determination of the data schemes and the metadata models that should be used for the correct recording and visualization of an element (as is documented by the relevant bibliographic review), no final decision has been made to impose the use of a specific data and metadata scheme/model of an element for each of the five ICH domains of the 2003 Convention. In the framework of the PhD thesis of the second author, a new and well-defined procedure for recording and visualizing ICH elements will be developed, so

that each time, the appropriate technologies and data/metadata standards will be selected to achieve the best possible results.

Additionally, as we have already mentioned on UNESCO's website, there are some videos of ICH elements that are not available anymore due to their copyrights. Although the 2003 UNESCO Convention for the Safeguarding of ICH provides appropriate terms and conditions regarding the copyright of photos and videos that are being attached to ICH elements, nevertheless, in this research, legal ambiguities and loopholes have been identified that ultimately allowed those attachments to be uploaded online. Thus, there is an urgent need to institute higher/more strict rules regarding the copyrights of those attachments and at the same time, UNESCO should determine relevant penalties for applicants' non-compliance with those new rules.

In addition, the Convention dictates in article 29 that the State Parties submit reports to the Commission on the legislative, regulatory, and other measures taken not only for safeguarding ICH in their territory, but also for updating the status of the elements that have already been registered in the Representative Lists. Although the UNESCO website provides an overview regarding the submission of periodic reports and each State Party's deadlines [60], it is observed that a lot of those reports have never been updated or do not contain the final filing date of the next required periodic reports. There should be a legal commitment with predefined penalties in cases where, after registration, the State Party does not report periodically to the Committee, the status of the item, the effectiveness of the safeguards that it has implemented, and the challenges it has faced during the whole process. The reader of this article must understand that among the means of ensuring the survival of an ICH element, is the update of the data that have already been recorded and visualized, by using new technologies which can more easily and efficiently imprint these data.

The fact that almost 50TB of data regarding ICH of Ukraine were in danger of being lost due to the threat of war, had probably not been foreseen by the decision makers in Ukraine, but even if it had been included in their studies of risk analysis, it was the way of storing the data (centralized local storage) which made the implementation of any countermeasures very difficult in such a threat. For this reason, the direct use of a different way of storing these data is required. Due to the fact that other countries are also following the exact same way of importing and storing the elements of their ICH in local repositories, the use of BT is being proposed. By creating a decentralized network between cultural heritage stakeholders all over the world and by developing a new dApp, the data of the world's ICH are going to be stored in a public ledger where each participant/miner of this network is forced to keep a full copy of the data or at least to keep a copy of the data of its neighbor's ICH data.

At this point, it is expected that the reader of this article might be slightly puzzled about whether it is possible to develop such a platform, which will support an ecosystem of members/users from different countries (State Parties of UNESCO) with different cultures, different possibilities in the use and installation of new technological solutions, etc. Of course, at a European level, this need has already been recorded. In fact, in October 2022, a funded project called Atlantis [61] started with a duration of 3 years (grant agreement No. 101073909), in which stakeholders from the private and public sectors from several European countries, including the first author of this article, participate in the development of an ecosystem that will operate under a single application (not a dApp one) which will be developed with the appropriate safety mechanisms to address systemic risks and will. Therefore, the first step, the recognition of the need for such an application, has not only been identified by the stakeholders but there is also a high possibility of funding such an attempt (the creation of the suggested dApp) by the European Union.

Undoubtedly, several problems are expected to arise during such a venture. For instance, the correct selection of members for the project team, the determination of the data that must be stored in each case of the five ICH domains, the data scheme, and the metadata model that researchers must use for the process of the recording and visualization of an

element, the determination of the control procedure on whether the data of a new element meet the predefined conditions/rules, also the development of these rules, the choice of the best possible consensus mechanism or the creation of a new one always respecting the environment (GHG emissions), and many others. At the same time, the use of new technologies is going to lead to new risks (systemic risks) which must be addressed using appropriate countermeasures, so as not to lead to a domino of ecosystem malfunctions.

Through the pages of this article, the authors approached the procedure of creating this new ecosystem and the development of a new decentralized application by using systems thinking to propose some possible solutions to the risks that may arise. In addition, through a systemic approach, the procedure to be followed has been explained, in as much detail as possible, to minimize the tendency of systems to resist the changes in their environment (E).

**Author Contributions:** Conceptualization, N.Z. and P.C.; methodology, N.Z. and P.C.; validation, N.Z. and P.C.; formal analysis, N.Z. and P.C.; investigation, N.Z. and P.C.; resources, N.Z. and P.C.; data curation, N.Z. and P.C.; writing—original draft preparation, N.Z., P.C. and N.A.; writing—review and editing, N.Z., P.C. and N.A.; visualization, N.Z. and P.C.; supervision, N.A.; project administration, N.A. All authors have read and agreed to the published version of the manuscript.

**Funding:** This research received no external funding.

**Informed Consent Statement:** Not applicable.

**Data Availability Statement:** The extracted results from the comparative study of the literature have not been posted online nor will they be available to the readers of this article because they are part of the Ph.D. thesis of the second author. Consequently, these data will be available after the submission, presentation and approval of her Ph.D. thesis.

**Acknowledgments:** The first two authors would like to thank the third author, Assimakopoulos Nikitas, for his guidance and for his continuous support throughout their research for their PhDs.

**Conflicts of Interest:** The authors declare no conflict of interest.

## Note

1   Miners are responsible for verifying transactions (applications for storing new data of ICH) and adding them to the blockchain. Mining nodes keep a copy of the entire blockchain and relay transactions.

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
