# Peer review of "Can UNESCO Use Blockchain to Ensure the Intangible Cultural Heritage of Humanity? A Systemic Approach That Explains the Why, How, and Difficulties of Such a Venture"

_heritage, doi:10.3390/heritage6030171_

Round 1

Reviewer 1 Report

Dear Authors,

the article discusses a very relevant and original topic. 

The systemic methodological approach is appreciated. 

It is only recommended to revise the conclusions and implement them with future research perspectives. 

Author Response

Point 1: It is only recommended to revise the conclusions and implement them with future research perspectives.

6. Conclusions

In the beginning, we talked about the vision of UNESCO and the criteria it has established in order to rescue an element of the ICH of humanity. We support that the nomination of an element of the ICH of humanity should be evaluated both on a socio-political economic level and also on a level of content completeness and technical specifications. Studying the files of the elements of the ICH of humanity, one is struck by how unequal the distribution of elements is in UNESCO’s lists during the consideration period (2008-2022). More specifically, the Representative List of the Intangible Cultural Heritage of Humanity includes 599 elements, the List of Intangible Cultural Heritage in Need of Urgent Safeguarding includes 76 elements and the Register of Good Safeguarding Practices includes only 33 elements [3]. What is more is the fact that in recent years humanity has faced a series of war conflicts (Ukraine, Iraq, etc.), natural disasters (like covid-19 pandemic, earthquakes, hurricanes), and thus in our opinion priority should be given to the List of Intangible Cultural Heritage in Need of Urgent Safeguarding. It is observed that the number of items registered in this list during the years 2008-2022 shows initially a decrease and stability afterward, although it should follow an increasing course. Emphasis should also be given to the Register of Good Safeguarding Practices, in which the average amount ranges between one (1) to five (5) elements per year (very low interest).

However, based on our research, both on its website and in UNESCO’s meeting minutes, it was found that there are no clear terms/rules regarding the technical specifications of the data that substantiate an element. It is self-evident that the lack of clear technical specifications creates confusion for stakeholders who apply for the induction of a new element of ICH, but also for their evaluators. Although in the past several projects have been funded regarding the determination of the data schemes and the metadata models that should be used for the correct record and visualization of an element (as it is documented by the relevant bibliographic review), no final decision has been made to impose the use of a specific data & metadata scheme/model of an element for each of the five (5) ICH domains of the 2003 Convention. In the framework of the Ph.D. thesis of the second author, a new and well-defined procedure for recording and visualizing ICH elements is going to be developed, so that each time the appropriate technologies and data/metadata standards are going to be selected to achieve the best possible results.

Additionally, as we have already mentioned on UNESCO’s website, there are some videos of ICH elements that are not available anymore due to their copyrights. Although the 2003 Convention for the Safeguarding of the ICH of UNESCO provides appropriate terms and conditions regarding the copyright of photos and videos that are being attached to ICH elements, nevertheless in this research legal ambiguities and loopholes have been identified that ultimately allowed those attachments to be uploaded online. Thus, there is an urgent need to institute higher/more strict rules regarding the copyrights of those attachments and at the same time, UNESCO should determine relevant penalties for applicants' non-compliance with those new rules.

Also, the Convention dictates in article 29 that the State Parties submit reports to the Commission on the legislative, regulatory, and other measures taken not only for safeguarding the ICH in their territory, but also for updating the status of the elements that have been already registered in the Representative Lists. Although the UNESCO website provides an overview regarding the submission of periodic reports and each State Member's deadlines [60], it is observed that a lot of those reports have never been updated or do not contain the final filing date of the next required periodic reports. There should be a legal commitment with predefined penalties in cases where, after registration, the State Party does not report periodically to the Committee, the status of the item, the effectiveness of the safeguards that it has implemented, and the challenges it has faced during the whole process. The reader of this article must understand that among the means of ensuring the survival of an ICH element is the update of the data that have already been recorded and visualized, by using new technologies which can more easily and efficiently imprint this data.

The fact that almost 50TB of data regarding the ICH of Ukraine got in danger to be lost due to the threat of war, probably had not been foreseen by the decision makers of Ukraine, but even if it had been included in their studies of Risk Analysis, it was the way of storing the data (centralized local storage) which made the implementation of any countermeasures very difficult in such a threat. For this reason, the direct use of a different way of storing this data is required. Due to the fact that other countries are also following the exact same way of importing and storing the elements of their ICH in local repositories, the use of BT is being proposed. By creating a Decentralized Network between cultural heritage stakeholders all over the world and by developing a new DApp, the data of the world ICH is going to be stored in a Public Ledger where each participant - miner of this network is forced to keep a full copy of the data or at least to keep a copy of the data of its neighbor’s ICH data.

At this point, it is expected that the reader of this article might be slightly puzzled about whether it is possible to develop such a platform, which will support an Ecosystem of member-users from different countries (State Parties of UNESCO) with different cultures, different possibilities in the use and installation of new technological solutions, etc. Of course, at a European level, this need has already been recorded. In fact, in October 2022 a funded project called Atlantis [61] started with a duration of 3 years (grant agreement No.101073909), in which stakeholders from the private and public sectors from several European countries, including the first author of this article, participate in the development of an Ecosystem that will operate under a single application (not a DApp one) which is going to be developed with the appropriate safety mechanisms to address systemic risks and will. So, the first step, the recognition of the need for such an application, not only has been identified by the stakeholders but there is also a high possibility of funding such an attempt (the creation of the suggested DApp) by the European Union.

Undoubtedly, several problems are expected to arise during such a venture. For instance, the correct selection of members for the Project Team, the determination of the data that must be stored in each case of the five (5) ICH domains, the data scheme, and the metadata model that researchers must use for the process of the record and the visualization of an element, the determination of the control procedure on whether the data of a new element meets the predefined conditions-rules, also the development of these rules, the choice of the best possible consensus mechanism or the creation of a new one always respecting the environment (GHG Emissions), and many others. At the same time, the use of new technologies is going to lead to new risks (Systemic Risks) which must be addressed using appropriate countermeasures, so as not to lead to a domino of Ecosystem malfunctions.

Through the pages of this article, the authors approached the procedure of creating this new Ecosystem and the development of a new Decentralized Application by using Systems Thinking to propose some possible solutions to the risks that may arise. Also, through a Systemic Approach, the procedure to be followed has been explained, in as much detail as possible, to minimize the tendency of Systems to resist the changes in their Environment (E).

Reviewer 2 Report

The article is well written and the research contributes to the field. I would recommend the authors to improve the introduction section by better specifying the aim of the research. 

Authors might want to specify the ammount of data they have searched at (and how) at the UNESCO database.

Finally, authors might add a discussion section to better define their results.

Author Response

Point 1:  I would recommend the authors to improve the introduction section by better specifying the aim of the research. 

1. Introduction (lines 98-105)

Therefore, the main purpose of the present paper is: (i) to record the need for a well-defined process by UNESCO, which must be followed worldwide, (2) to compare some of the metadata models that have been presented in the past few years in order to find the common data that must be, at least, recorded (iii) to specify why Blockchain Technologies (BT) are better (from a perspective of data security) for storing the global ICH data and to present a Decentralized application (DApp) which must be used by UNESCO (a multi-layer architecture of this DApp is going to be presented which has been deployed using a Systemic approach).

Point 2: Authors might want to specify the amount of data they have searched at (and how) at the UNESCO database.

1. Introduction  (lines 80-91)

Although we have seen many countries like Greece, China, France, Croatia, Mauritania, Iran, Brazil, United Arab Emirates, Uganda, Morocco, Egypt, and many more which have successfully followed the procedure to add a new element to their Intangible Cultural Heritage, it is worth highlighting that only 76 of those 708 elements, belonging in 40 different countries, have been added since 2009 in the List of Intangible Cultural Heritage in Need of Urgent Safeguarding [6].

More specifically, 12 elements have been added in 2009, 4 elements in 2010, 10 elements in 2011, 4 elements in 2012, 4 elements in 2013, 3 elements in 2014, 5 elements in 2015, 4 elements in 2016, 6 elements in 2017, 7 elements in 2018, 5 elements in 2019, 3 elements in 2020, 4 elements in 2021 and 5 elements in 2022. So, we can safely conclude that in the past few years, a decreasing trend in the effort to record new elements into the List of Intangible Cultural Heritage in Need of Urgent Safeguarding has been observed.

2. Data of ICH & Standards of Metadata. Literature review (lines 219-230)

At this point, in order to track the way UNESCO stores the data in its database, we visited the website https://ich.unesco.org/dive/domain [16] and from the picture in the center of the screen (Figure 1) we chose the domain “Traditional Craftsmanship”. Then, we chose the bullet with the name “Tango” which popped up a new window with all the data of this specific ICH element. We traced that the data are stored locally and the video is stored in the public FTP server of YouTube (we must mention that this video is unavailable due to its copy rights), but there is an interconnection between them. Also, the photos of this element are available by clicking on the "next" button and if we select the hyperlink "See all the information available on this element" we can confirm that the photos are stored locally. We repeated the same process for almost 2/3 of the data (in each one of the 5 domains) and we were led to exactly the same conclusions about the way that those data are being stored. 

Point 3: Authors might add a discussion section to better define their results.

6. Conclusions 

(lines 883-934)

We support that the nomination of an element of the ICH of humanity should be evaluated both on a socio-political economic level and also on a level of content completeness and technical specifications. Studying the files of the elements of the ICH of humanity, one is struck by how unequal the distribution of elements is in UNESCO’s lists during the consideration period (2008-2022). More specifically, the Representative List of the Intangible Cultural Heritage of Humanity includes 599 elements, the List of Intangible Cultural Heritage in Need of Urgent Safeguarding includes 76 elements and the Register of Good Safeguarding Practices includes only 33 elements [3]. What is more is the fact that in recent years humanity has faced a series of war conflicts (Ukraine, Iraq, etc.), natural disasters (like covid-19 pandemic, earthquakes, hurricanes), and thus in our opinion priority should be given to the List of Intangible Cultural Heritage in Need of Urgent Safeguarding. It is observed that the number of items registered in this list during the years 2008-2022 shows initially a decrease and stability afterward, although it should follow an increasing course. Emphasis should also be given to the Register of Good Safeguarding Practices, in which the average amount ranges between one (1) to five (5) elements per year (very low interest).

However, based on our research, both on its website and in UNESCO’s meeting minutes, it was found that there are no clear terms/rules regarding the technical specifications of the data that substantiate an element. It is self-evident that the lack of clear technical specifications creates confusion for stakeholders who apply for the induction of a new element of ICH, but also for their evaluators. Although in the past several projects have been funded regarding the determination of the data schemes and the metadata models that should be used for the correct record and visualization of an element (as it is documented by the relevant bibliographic review), no final decision has been made to impose the use of a specific data & metadata scheme/model of an element for each of the five (5) ICH domains of the 2003 Convention. In the framework of the Ph.D. thesis of the second author, a new and well-defined procedure for recording and visualizing ICH elements is going to be developed, so that each time the appropriate technologies and data/metadata standards are going to be selected to achieve the best possible results.

Additionally, as we have already mentioned on UNESCO’s website, there are some videos of ICH elements that are not available anymore due to their copyrights. Although the 2003 Convention for the Safeguarding of the ICH of UNESCO provides appropriate terms and conditions regarding the copyright of photos and videos that are being attached to ICH elements, nevertheless in this research legal ambiguities and loopholes have been identified that ultimately allowed those attachments to be uploaded online. Thus, there is an urgent need to institute higher/more strict rules regarding the copyrights of those attachments and at the same time, UNESCO should determine relevant penalties for applicants' non-compliance with those new rules.

Also, the Convention dictates in article 29 that the State Parties submit reports to the Commission on the legislative, regulatory, and other measures taken not only for safeguarding the ICH in their territory, but also for updating the status of the elements that have been already registered in the Representative Lists. Although the UNESCO website provides an overview regarding the submission of periodic reports and each State Member's deadlines [60], it is observed that a lot of those reports have never been updated or do not contain the final filing date of the next required periodic reports. There should be a legal commitment with predefined penalties in cases where, after registration, the State Party does not report periodically to the Committee, the status of the item, the effectiveness of the safeguards that it has implemented, and the challenges it has faced during the whole process. The reader of this article must understand that among the means of ensuring the survival of an ICH element is the update of the data that have already been recorded and visualized, by using new technologies which can more easily and efficiently imprint this data.

(lines 947-959)

At this point, it is expected that the reader of this article might be slightly puzzled about whether it is possible to develop such a platform, which will support an Ecosystem of member-users from different countries (State Parties of UNESCO) with different cultures, different possibilities in the use and installation of new technological solutions, etc. Of course, at a European level, this need has already been recorded. In fact, in October 2022 a funded project called Atlantis [61] started with a duration of 3 years (grant agreement No.101073909), in which stakeholders from the private and public sectors from several European countries, including the first author of this article, participate in the development of an Ecosystem that will operate under a single application (not a DApp one) which is going to be developed with the appropriate safety mechanisms to address systemic risks and will. So, the first step, the recognition of the need for such an application, not only has been identified by the stakeholders but there is also a high possibility of funding such an attempt (the creation of the suggested DApp) by the European Union.

Reviewer 3 Report

It is a good paper on timely and relevant issues that falls short on two points:

1) the part explaining how blockchain works could be (or perhaps should be) shortened. The technology is well-known, and even people with non-technical backgrounds are familiar. So, making this part much shorter and aimed at readers who have no idea how blockchain works and would like a no-nonsense introduction would be enough.

2) the legal component is missing. I absolutely concur with the idea of digitalizing intangible cultural heritage and applying blockchain technology in this area. Of course, it can be applied to initiatives like SUCHO.ORG i.e. Saving Ukrainian Heritage Online, its worldwide application should be in line with UNESCO 2003 convention and should also fit somehow in national legislations. The million dollar question, and perhaps the most important in this area, is how to implement the proposed project and what legal actions will be required?

Author Response

Point 1: Making the Blockchain part much shorter and aimed at readers who have no idea how blockchain works and would like a no-nonsense introduction would be enough.

Please see the review of chapter 5. How & Why Blockchain Technologies must be used for storing the Data of the Intangible Cultural Heritage of Humanity in the attachment file.

Point 2: The legal component is missing. How to implement the proposed project and what legal actions will be required?

(lines 908-934)

In the framework of the Ph.D. thesis of the second author, a new and well-defined procedure for recording and visualizing ICH elements is going to be developed, so that each time the appropriate technologies and data/metadata standards are going to be selected to achieve the best possible results.

Additionally, as we have already mentioned on UNESCO’s website, there are some videos of ICH elements that are not available anymore due to their copyrights. Although the 2003 Convention for the Safeguarding of the ICH of UNESCO provides appropriate terms and conditions regarding the copyright of photos and videos that are being attached to ICH elements, nevertheless in this research legal ambiguities and loopholes have been identified that ultimately allowed those attachments to be uploaded online. Thus, there is an urgent need to institute higher/more strict rules regarding the copyrights of those attachments and at the same time, UNESCO should determine relevant penalties for applicants' non-compliance with those new rules.

Also, the Convention dictates in article 29 that the State Parties submit reports to the Commission on the legislative, regulatory, and other measures taken not only for safeguarding the ICH in their territory, but also for updating the status of the elements that have been already registered in the Representative Lists. Although the UNESCO website provides an overview regarding the submission of periodic reports and each State Member's deadlines [60], it is observed that a lot of those reports have never been updated or do not contain the final filing date of the next required periodic reports. There should be a legal commitment with predefined penalties in cases where, after registration, the State Party does not report periodically to the Committee, the status of the item, the effectiveness of the safeguards that it has implemented, and the challenges it has faced during the whole process. The reader of this article must understand that among the means of ensuring the survival of an ICH element is the update of the data that have already been recorded and visualized, by using new technologies which can more easily and efficiently imprint this data.

(lines 947-959)

At this point, it is expected that the reader of this article might be slightly puzzled about whether it is possible to develop such a platform, which will support an Ecosystem of member-users from different countries (State Parties of UNESCO) with different cultures, different possibilities in the use and installation of new technological solutions, etc. Of course, at a European level, this need has already been recorded. In fact, in October 2022 a funded project called Atlantis [61] started with a duration of 3 years (grant agreement No.101073909), in which stakeholders from the private and public sectors from several European countries, including the first author of this article, participate in the development of an Ecosystem that will operate under a single application (not a DApp one) which is going to be developed with the appropriate safety mechanisms to address systemic risks and will. So, the first step, the recognition of the need for such an application, not only has been identified by the stakeholders but there is also a high possibility of funding such an attempt (the creation of the suggested DApp) by the European Union.
